# 3D Bio-Printing of CS/Gel/HA/Gr Hybrid Osteochondral Scaffolds

**DOI:** 10.3390/polym11101601

**Published:** 2019-09-30

**Authors:** Xueyan Hu, Yuan Man, Wenfang Li, Liying Li, Jie Xu, Roxanne Parungao, Yiwei Wang, Shuangshuang Zheng, Yi Nie, Tianqing Liu, Kedong Song

**Affiliations:** 1State Key Laboratory of Fine Chemicals, Dalian R&D Center for Stem Cell and Tissue Engineering, Dalian University of Technology, Dalian 116024, China; Huxueyan@mail.dlut.edu.cn (X.H.); 975202979@mail.dlut.edu.cn (Y.M.); liwenfang@mail.dlut.edu.cn (W.L.); liyingli@mail.dlut.edu.cn (L.L.); xujie172@mail.dlut.edu.cn (J.X.); 2Burns Research Group, ANZAC Research Institute, University of Sydney, Concord, NSW 2139, Australia; rpar4161@uni.sydney.edu.au (R.P.); yiweiwang@anzac.edu.au (Y.W.); 3Zhengzhou Institute of Emerging Industrial Technology, Zhengzhou 450000, China; sszheng@ipezz.ac.cn; 4Key Laboratory of Green Process and Engineering, Institute of Process Engineering, Chinese Academy of Sciences, Beijing 100190, China

**Keywords:** 3D printing, bio-ink, graphene, chitosan/gelatin/hyaluronic acid, cartilage repair

## Abstract

Cartilage is an important tissue contributing to the structure and function of support and protection in the human body. There are many challenges for tissue cartilage repair. However, 3D bio-printing of osteochondral scaffolds provides a promising solution. This study involved preparing bio-inks with different proportions of chitosan (Cs), Gelatin (Gel), and Hyaluronic acid (HA). The rheological properties of each bio-ink was used to identify the optimal bio-ink for printing. To improve the mechanical properties of the bio-scaffold, Graphene (GR) with a mass ratio of 0.024, 0.06, and 0.1% was doped in the bio-ink. Bio-scaffolds were prepared using 3D printing technology. The mechanical strength, water absorption rate, porosity, and degradation rate of the bio-scaffolds were compared to select the most suitable scaffold to support the proliferation and differentiation of cells. P3 Bone mesenchymal stem cells (BMSCs) were inoculated onto the bio-scaffolds to study the biocompatibility of the scaffolds. The results of SEM showed that the Cs/Gel/HA scaffolds with a GR content of 0, 0.024, 0.06, and 0.1% had a good three-dimensional porous structure and interpenetrating pores, and a porosity of more than 80%. GR was evenly distributed on the scaffold as observed by energy spectrum analyzer and polarizing microscope. With increasing GR content, the mechanical strength of the scaffold was enhanced, and pore walls became thicker and smoother. BMSCs were inoculated on the different scaffolds. The cells distributed and extended well on Cs/Gel/HA/GR scaffolds. Compared to traditional methods in tissue-engineering, this technique displays important advantages in simulating natural cartilage with the ability to finely control the mechanical and chemical properties of the scaffold to support cell distribution and proliferation for tissue repair.

## 1. Introduction

Cartilage is an important tissue in humans and animals, contributing to the structure and function of the ear, nose, intervertebral disc meniscus, and interosseous joint [1]. However, when cartilage is damaged, it has a limited capacity for self-repair as it lacks nerve, lymphatic, and blood supply [2,3,4,5]. As cartilage tissue can be subjected to multiple injuries and involves complex therapeutic pathology, repairing cartilage to restore normal structure and function is one of the most challenging areas in orthopedic research and sports medicine. Tissue engineering is an interdisciplinary field that provides a prospective alternative platform to implant chondrocytes [6,7,8,9,10].

3D bio-printing is the “additive manufacturing” of objects, as objects are constructed by additive deposition of materials [11,12]. Bio-printing represents a potential tool for organ regeneration in the future, and has broad potential applications in tissue engineering [13]. Compared to traditional methods, this technique shows important advantages in the development of cartilage tissue engineering, providing the opportunity to finely control the mechanical and chemical properties that can influence cell distribution and behavior [14,15]. This opens many new perspectives [16,17,18] for the development of complex structures (osteochondral septum), different types of cartilage (transparent fibers), and personalized medicine based on the needs of specific patients [19,20,21]. 

Natural materials commonly used in cartilage repair are alginate, chitosan (Cs), collagen, and hyaluronic acid (HA) [22,23]. Cs is a natural polysaccharide extracted from crustacean shells [24]. Cs contains glucosamine and HA, which are the basic components of natural cartilage [25,26,27]. Therefore, Cs is widely used in cartilage tissue engineering [28,29]. Recent studies have shown that Cs-HA hydrogel promotes the healing of cartilage in rabbit models of injury [30]. Collagen is the main component of the extracellular matrix (ECM) of chondrocytes, and collagen gel has been widely used and has positive outcomes as the matrix for the replacement of articular cartilage. Injectable type II collagen gel has been used to treat full-thickness articular cartilage defects [31]. Clinical studies have shown that collagen gel can be used to replace cartilage and subchondral bone [32]. HA is the main component of native cartilage. HA [33,34] is an anionic non-sulfated glycosaminoglycan that exists in cartilage ECM and synovial fluid, provides a stem cell niche, and supports cell adhesion through its expression of surface receptors such as CD44 [35,36]. Similar to Cs scaffolds, HA is the most widely used in cartilage tissue engineering scaffolds.

Carbon-based nanomaterials have been widely studied for biomedical applications [37,38]. Graphene (GR) is a member of the family of carbon-derived nanomaterials [39], and owing to its excellent electrical conductivity, high tensile strength, and simple functional group bonding, it has multifunctional applications in the biomedical field. GR has been used to improve the surface characteristics of biomaterials and bone regeneration scaffolds [40,41,42], and has been used in drug delivery and scaffold [43,44,45]. GR biomaterials have been used for the chondrogenic differentiation of mesenchymal stem cells (MSCs) [46,47]. Biomaterials which consist of chondroitin methacrylate sulfate, 2-hydroxy-1-(4-(hydroxyethoxy) phenyl)-2-methyl-1-propanone and GR have been prepared for cartilage construction [48,49].

The traditional preparation technology of cartilage tissue engineering scaffolds has many shortcomings such as a limited shape of scaffolds, difficulty in controlling pore size, and insufficient mechanical strength. 3D printing technology can overcome the limitations of traditional stent manufacturing methods in terms of shape and process consistency and realize the preparation of high-precision scaffolds. As articular cartilage is expected to have a smooth surface and the ability to withstand large mechanical loads, a number of materials are used to accurately simulate the structure and function of natural articular cartilage [50]. In this study, Cs, Gel, and HA were used to compound, each providing properties to support articular formation. Cs, Gel, and HA have low mechanical properties; however, adding GR to these materials is expected to enhance the mechanical properties [51] and biocompatibility of these scaffolds.

## 2. Materials and Methods

### 2.1. Experimental Design

An overview of the experimental design is illustrated in Figure 1. Cs solution was prepared by dissolving Cs powder in 2% (*v*/*v*) dilute acetic solution. Gel solution and HA solution were obtained by dissolving Gel and HA in double distilled water. Using a magnetic agitator, Cs, Gel, and HA solutions were mixed together for 4 h at 45 °C. The bio-ink was finally prepared with the addition of GR to the mixture of Cs, Gel, and HA solutions. Since the scaffolds need to be crosslinked with NHS/MES/EDC, and the optimal temperature of EDC was at pH of 4–6, so the working pH of gelatin has been adjusted in the range of 4–6. The rheological properties of each bio-ink preparation was tested to identify the optimal preparation for printing. The Cs/Gels/HA/GR scaffolds were printed using a 3D printer (Hangzhou Jienuofei Biotechnology Co., Ltd. Hangzhou, China), and the physical and chemical properties of bio-scaffolds were tested. BMSCs were isolated and cultured from rats, and after the third passage, were seeded onto the Cs/Gels/HA/GR scaffolds before the biocompatibility of Cs/Gels/HA/GR scaffolds were tested.

### 2.2. Experimental Materials

Cs (Degree of deacetylation > 90%), Gel, Beijing Coolaber Technology Co., Ltd. (Beijing, China); glacial acetic acid, Tianjin Damao Chemical Reagent Factory (Tianjin, China); disodium hydrogen phosphate, Tianjin Komiou Chemical Reagent Co., Ltd. (Tianjin, China); HA, Beijing Solarbio Technology Co., Ltd. (Beijing, China); GR, Institute of Chemistry, Chinese Academy of Sciences; 3D printing (Pro), Hangzhou Jienuofei Biotechnology Co., Ltd. (Hangzhou, China); Rheometer (MCR302), Guangzhou Yice Instrument Co., Ltd. (Guangzhou, China); Freeze Dryer (LGJ-10N), Beijing Yaxingyike Technology Development Co., Ltd. (Beijing, China); EDC/NHS/MES, Energy Chemical (Shanghai, China).

### 2.3. Preparation of Biological Ink

Cs solution was prepared by dissolving Cs powder in 2% (*v*/*v*) dilute acetic solution. Gel solution and HA solution were obtained by dissolving Gel and HA in double-steamed water. The final concentration and configuration ratio of each solution are shown in Table 1. Using a magnetic agitator, Cs, Gel, and HA solutions were mixed together for 4 h at 45 °C. When the mixture was evenly mixed, the bio-ink was prepared by standing defoaming. The bio-ink was prepared by mixing Cs/Gel/HA in a volume ratio of 1:8:0.02 with 0, 0.024, 0.06 and 1% GR. This prepared bio-ink was then placed in a 3D printer to print the bio-scaffolds. The conditions for printing are shown in Table 2. After printing and molding, the scaffold is pre-frozen for 12 h in −20 °C and then transferred to the freeze-dryer, where it is pre-frozen for 1 h under −71 °C and vacuum-dried for 12 h. The bio-scaffolds were cross-linked with EDC/NHS/MES for 6 h, and were then incubated in 0.1 mol/L Na2HPO4 solution for 2 h at room temperature. The scaffolds were then rinsed repeatedly with deionized water to remove any residual cross-linking agent and acetic acid solution during the preparation of the scaffold. The prepared Cs/Gel/HA and Cs/Gel/HA/GR scaffold were then stored for analysis.

### 2.4. Rheological Properties of Bio-Ink

Each prepared bio-ink was placed in the rheometer (Guangzhou Yice Instrument Co., Ltd. Guangzhou, China) to determine the fluid characteristics of each preparation. To determine the relationship between shear rate and viscosity, the measuring position was set to 0.1 mm, and the shear rate ranged from 0.1 to 1000 s^−1^. To measure the relationship between temperature and viscosity, the frequency was fixed at 1Hz, and the rising and cooling speed was set to 5 °C/min. The relationship between strain and modulus was tested at a frequency of 1Hz, and a strain range between 0.01–10%. The relationship between frequency and modulus was measured, and frequency between 0.1–100 Hz. Under constant shear rate, and the rising and cooling speed at 5 °C/min, the relationship between temperature and modulus was also determined.

### 2.5. Scanning Electron Microscopic and Energy Spectrum of Bio-Scaffold

The bio-scaffolds were cut into 8 mm × 8 mm × 2 mm pieces before the samples were purged with nitrogen, and subsequently sprayed with gold, to observe the microstructure and pore structure of the scaffold using a tungsten filament scanning electron microscope (FEI Company, Hillsboro, OR, USA). The accelerating voltage used was 30 kV and the limit resolution of the tungsten filament scanning electron microscope was 3 nm. The bio-scaffolds were selected for EDS semi-quantitative analysis, and the element types and contents of the stent were analyzed.

### 2.6. Polarizing Microscope of Bio-Scaffold

The bio-scaffolds were cut into 8 mm × 8 mm × 2 mm pieces before being placed in a chamber, and the focal length adjusted until a clear image of the sample could be obtained. And the images are generated using the LAS V4.11 software.

### 2.7. Water Absorption of Bio-Scaffold

To determine the water absorption of the prepared bio-scaffolds, the dry weight of each scaffold was first recorded (*m*_0_). The bio-scaffolds were then immersed in distilled water at room temperature for 3, 6, 24, 48, or 96 h. Bio-scaffolds were removed and placed on dehydrating filter paper, and any excess moisture was gently wiped away on the surface. The wet weight of each bio-scaffold was then recorded (*m_t_*) to calculate the water absorption rate (*P* (%)) using the following equation:(1)$$P(\%)=\frac{m_{t}-m_{0}}{m_{0}}\times 100\%$$

### 2.8. Porosity of Bio-Scaffold

The bio-scaffolds were immersed in anhydrous ethanol with a volume of *V*_1_ and placed in a vacuum drying oven. After the bio-scaffolds were completely infiltrated with anhydrous ethanol, the total volume between bio-scaffolds and anhydrous ethanol was *V*_2_. The bio-scaffolds were then removed, and the remaining volume of ethanol was defined as *V*_3_. The volume of the bio-scaffold is *V*_2_ − *V*_1_. The volume occupied by bio-scaffold pores is *V*_1_ − *V*_3_. The apparent volume of the bio-scaffold is the sum of the volume of the bio-scaffold and the volume occupied by the bio-scaffold pores, *V* = (*V*_2_ − *V*_1_) + (*V*_1_ − *V*_3_) = *V*_2_ − *V*_3_. The porosity *A* (%) of each bio-scaffolds was then calculated using the following equation:(2)$$A(\%)=\frac{V_{1}-V_{3}}{V_{2}-V_{3}}\times 100\%$$

### 2.9. Degradation Rate of Bio-Scaffold

To determine the degradation rate of the prepared bio-scaffolds, the dry weight of each bio-scaffold was recorded (*m*_0_). The bio-scaffolds were then placed in lysozyme degradation solution with a concentration of 12 ug/mL. The bio-scaffolds were removed from the solution and allowed to dry every 3 d, and the dry weight subsequently recorded (*m*_1_) over a total of two weeks. The degradation rate D (%) of each bio-scaffolds was calculated using the following equation:(3)$$D(\%)=\frac{\left(m_{0}-m_{1}\right)}{m_{0}}\times 100\%$$

### 2.10. Mechanical Properties of Bio-Scaffold

The compression modulus of Cs/Gel/HA bio-scaffolds with 0, 0.024, 0.06, 0.1% GR was measured using a universal experimental machine. The upper and lower surface of the bio-scaffold was cut into the size of a small cube (8 mm × 8 mm × 5 mm). The displacement control loading speed of the universal test machine is 1 mm/min. The compression modulus *E* (%) of each bio-scaffolds was calculated using the following equation:(4)$$E=\frac{\varepsilon }{\sigma }=\frac{10(L_{2}-L_{1})/S}{(D_{2}-D_{1})/h}$$

In the formula, *D*_1_ and *D*_2_ are the displacement of the bio-scaffold. *S* and h are the cross-section area and height of the bio-scaffold, respectively. *L*_1_ and *L*_2_ are the pressure loads before and after the beginning of the linear segment.

### 2.11. Biocompatibility of Bio-Scaffold

Cs/Gel/HA/GR scaffolds were soaked in 75% alcohol, and the scaffolds were exposed to ultraviolet light for disinfection and sterilization. The bio-scaffolds were then washed with PBS to remove any excess alcohol. P3 BMSCs were inoculated into scaffolds with a density of 1.0 × 10^8^ cells/mL and cultured in 37 °C and 5% CO_2_ incubators.

Cell scaffold complexes were stained with calcein, Hoechst and PI after 1, 3, 5 and 7 d of culture. The bio-scaffolds were incubated for 30 min in the incubator and placed under fluorescence microscope to observe the growth of cells on the bio-scaffold.

Cell scaffold complexes were fixed with glutaraldehyde after 1, 3, 5 and 7 d of culture. The scaffolds were successively dehydrated with an aqueous solution containing 50, 70, 90 and 100% ethanol concentration, and were allowed to dry before being and sprayed with gold. The accelerating voltage used was 30 kV and the limit resolution of the tungsten filament scanning electron microscope was 3 nm.

### 2.12. Statistical Analysis

Data were presented as mean ± standard deviations. All data were statistically analyzed using Origin 9.0. *p* < 0.05, *p* < 0.01 and *p* < 0.001 were considered statistically significant. 

## 3. Results and Discussions

### 3.1. Rheological Properties of Bio-Ink

The relationship between viscosity and changes in shear rate was measured in various preparations of Cs:Gel:HA bio-ink (Figure 2a). Bio-ink prepared with a Cs:Gel ratio of 2:6, 1:7, and 1:8 demonstrated the phenomenon of shear thickening, with viscosity increasing with increased shear stress. Under these conditions, the intermolecular winding point increases, making the bio-ink nonoptimal for printing. However, when the total proportion of Cs and Gel is 6%, and incorporates HA, the bio-ink demonstrates the phenomenon of shear thinning. With increasing shear stress, points of intermolecular entanglement decrease, macromolecules disentangle and orient along the directional flow due to conformation changes under shear action. The rate of destruction of the entangled structure is greater than the rate of formation. As there are fewer points of entanglement between molecules in the bio-ink, the intermolecular force between the molecules decreases, decreasing viscosity. The bio-ink ratio of Cs:Gel = 1:5, Cs:Gel:HA = 1:8:0.02, and 1:8:0.06 have high viscosity at low shear rate. Therefore, these three solutions are more stable and less prone to sedimentation, making them more suitable for 3D bio printing.

The relationship between temperature and viscosity was also assessed for each bio-ink preparation (Figure 2b). In general, for each bio-ink preparation, viscosity decreased with an increase in temperature. However, for bio-inks with a Cs:Gel = 2:6, 1:7 and 1:8, the viscosity for these preparations were relatively higher compared to Cs:Gel = 2:4, 1:5 and Cs:Gel:HA = 1:8:0.02, 1:8:0.06, 1:8:0.1. Which is similar to the phenomenon of shear stress scanning. The bio-inks with a Cs:Gel = 2:6, 1:7 and 1:8, were relatively stable, had higher viscosity, but difficult to extrude. Bio-inks composed of equal ratios of Cs and Gel with a high content of gelatin, was reported to be have high viscosity, indicating that gelatin greatly influences the viscosity of the bio-ink. When comparing the viscosity of bio-inks with Cs:Gel = 1:8,Cs:Gel:HA = 1:8:0.02,1: 8:0.006 and 1:8:0.1, bio-inks doped with HA, had reduced viscosity under the same temperature conditions. In particular, the bio-ink with a ratio of Cs:Gel:HA = 1:8:0.02 maintained a low viscosity at 20 °C. These results suggest that incorporating HA has the potential to influence the characteristics of the bio-ink fluid, modifying the viscosity of the fluid, and influencing any shifts between shear thickening fluid and shear thinning fluid. It indicates that the impact on viscosity of bio-ink is HA > Gel > Cs.

The relationship between shear strain and modulus of storage and loss in different preparations of bio-ink were assessed (Figure 2c_1_,c_2_). When the storage and loss modulus no longer change with changes in strain, this is defined as the linear viscoelastic region of the material, and a region where it is impossible to damage the structure of the material. The larger the linear viscoelastic region, the more stable the material is. The linear viscoelastic zone of each proportional bio-ink is shown in Table 3. Cs:Gel:HA = 1:8:0.2 and Cs:Gel:HA = 1:8:0.1 have a small linear range. The ratio of Cs:Gel = 1:5 and Cs:Gel = 2:6 bio-inks have higher storage and loss modulus. The storage and loss modulus of Cs:Gel = 1:5 is higher than 100 Pa, and the storage and loss modulus of Cs:Gel = 2:6 are between 10–30 pa. The storage and loss modulus of bio-ink is: Cs:Gel = 1:5 > Cs:Gel = 2:6 > Cs:Gel:HA = 1:8:0.02 > Cs:Gel = 1:7 > Cs:Gel = 1:8 > Cs:Gel:HA = 1:8:0.1 > Cs:Gel = 2:4 > Cs:Gel:HA = 1:8:0.06.

Frequency sweep curves were produced for the various preparations of bio-ink (Figure 2d_1_,d_2_). The behavior of the prepared bio-inks was measured. Bio-inks for printing should be elastic and show solid-like behavior. The behavior of bio-inks with different ratios were assessed from low frequency to high frequency, and is summarized in Table 4. The storage and loss modulus of bio-ink is Cs:Gel = 2:6 > Cs:Gel:HA = 1:8:0.02 > Cs:Gel = 2:4, and is relatively higher when compared to the other preparations. Strain and frequency scanning of Cs:Gel:HA = 1:8:0.06 show that the storage and loss modulus are both the lowest, and the material is not very stable. Bio-inks that exhibit viscoelastic liquid behavior are not suitable for printing. Under low-frequency and high-frequency scanning, Cs:Gel = 2:4,Cs:Gel = 2:6,Cs:Gel = 1:5 and Cs:Gel:HA = 1:8:0.02 bio-inks all demonstrated solid-like behavior and were relatively stable, making them suitable for printing.

Similarly, the fluid properties of bio-inks with various proportions was assessed (Table 5). The fluid properties of Cs:Gel = 2:4 and Cs:Gel: HA = 1:8:0.02 of bio-inks make them a suitable choice for printing. The gel points of these two bio-ink preparations are shown in Figure 3a,b. Cs:Gel = 2:4 bio-ink has a gel point of 23 °C. When the temperature of the bio-ink is below 23 °C, the fluid behavior is close to solid. When the temperatures rise above 23 °C, the fluid behavior is close to liquid. However, the gel point of Cs:Gel:HA = 1:8:0.02 is 31 °C, so its structure is more stable and closer to room temperature, making it the most suitable bio-ink for printing.

### 3.2. Morphology of Bio-Ink

Selecting the appropriate needle diameter for bio-printing is important as it can influence the expansion of the bio-ink when it is extruded from the needle [52,53]. A number of needles with varying diameter sizes were tested to determine the optimal needle for printing. Needle diameters ranged from 0.11 mm to 0.51 mm, among which the 0.41 mm and 0.51 mm size ones were needle-type and conical needles (Figure 4A). It was important to note that the cartilage scaffold should have small pores at a micron scale. Furthermore, because the prepared bio-ink is thick, it will appear gelatinous at low temperatures. Due to the trace amounts of GR doped in the bio-ink, GR does not dissolve in water, and may settle in the syringe for too long, causing blockage in the needle. We found that 0.21 mm was the smallest diameter needle that could be used to print the scaffold structure, and worked well with the Cs:Gel:HA 1:8:0.02 bio-ink. Since this bio-ink preparation does not contain GR, the chance of any blockage was minimised. For the bio-ink preparations that contained GR, conical needles were used, allowing for the gradual extrusion of the bio-ink and preventing the accumulation of GR.

Bio-scaffolds were prepared as 8 mm × 8 mm × 5 mm pieces. A 3D scaffold printing model (Figure 4B) and the extrusion of bio-ink from the needle by the 3D printer (Figure 4C) is demonstrated. As the pore size of the scaffolds directly affects the proliferation and differentiation of cells, the optimal filling spacing was investigated (Figure 4D_1_–D_6_). After observing differences in filling space under different conditions, the 0.8 mm filling space was selected as the most appropriate to support cell proliferation and differentiation (Figure 4D_1_–D_6_). Scaffolds for the different bio-ink preparations were also printed (Figure 4E_1_–E_4_). These printed scaffolds had a uniform pore diameter, and that Cs/Gel/HA scaffolds were white, but with increasing GR content the scaffolds appeared black.

### 3.3. Physical and Chemical Properties of Bio-Scaffolds

The physical and chemical properties of the prepared bio-ink scaffolds were assessed to determine its suitability as a Cs:Gel:HA = 1:8:0.02 in cartilage repair. The surface morphology of the printed scaffolds from different bio-ink preparations was assessed using SEM (Figure 5a). The pore wall thickened with increasing GR content, but had no effect on surface structure. The diameter of large pores were measured to be 30–300 μm, and micropores at a range of 20–50 μm. The porous structure of these scaffolds mean that they can support the transfer of nutrients and removal of metabolized waste.

A polarizing microscope was used to observe whether GR was evenly distributed within the scaffolds (Figure 5b). It can be clearly observed that as GR content increased, the color of the scaffolds became darker. When the GR content was 0.1%, the scaffold was almost completely covered with GR, with the GR evenly distributed on the pore wall of the scaffold.

EDS semi-quantitative analysis of scaffolds was performed for all scaffolds printed using different bio-ink preparations (Figure 5c). A higher carbon content is indicative of a higher GR content in the scaffolds. The mass percentage of carbon element of CS/Gel/HA, CS/Gel/HA/0.024%GR, CS/Gel/HA/0.06%GR and CS/Gel/HA/0.1%GR, was measured to be 56.27%, 56.47%, 56.84%, 57.04%, respectively. These results highlight that GR has been successfully doped into the scaffolds.

The water absorption rate of the four scaffolds after 3, 6, 24, 48, and 96 h in PBS solution was measured (Figure 6a). As GR is a hydrophobic substance, scaffolds with GR have a reduced capacity to absorb water compared to scaffolds without any GR. The water absorption rate of CS/Gel/HA, CS/Gel/HA/0.024% GR, CS/Gel/HA/0.06% GR and CS/Gel/HA/0.01% GR scaffold was reported to be 872.5 ± 59.4%, 726.31 ± 14.54%, 735.04 ± 50.83% and 766.45 ± 15.72%, respectively. But the water absorption rate will increase when the GR content increases, because GR is associated with a firm interaction of Cs during the nucleation step, which makes the structure more hydrophilic. With the hydrophilicity increased, the diffusion of water molecules to the growing crystals will increase, which leads to the water absorption rate increasing. These results indicate that all scaffolds have good hydrophilicity and can meet the requirements of cell proliferation and adhesion on scaffolds.

The porosity of scaffolds printed using different bio-ink preparations were measured (Figure 6b). A higher scaffold porosity is optimal, allowing for more opportunities for cell penetration and nutrient exchange. All scaffolds were highly porous. The porosity of CS/Gel/HA, CS/Gel/HA/0.024% GR, CS/Gel/HA/0.06% GR and CS/Gel/HA/0.1% GR scaffolds were 87.37 ± 3.59%, 85.86 ± 3.60%, 84.58 ± 3.87%, and 86.05 ± 0.84%, respectively. The GR-doped scaffolds had a slightly lower porosity than the control scaffold, but no significant differences could be detected. Our SEM analysis demonstrate that GR content thickens the pore wall of the scaffold, but increasing GR added does not change the internal structure of the scaffold.

The degradation rate of prepared scaffolds were also determined (Figure 6c). The degradation rate decreased gradually in a scaffold with increased GR content. However, the degradation rate of the CS/Gel/HA/0.1% GR scaffolds was slightly higher, potentially owing to its high absorption rate and ability to take up more degradation fluid. After day 3, the degradation rate of all scaffolds showed a gradually increasing trend. In the first 3 d, the degradation rate was very fast for the CS/Gel/HA and CS/Gel/HA/0.1% GR scaffolds. This may be explained by the rough pore walls of CS/Gel/HA/0.1% GR, which increases the surface area and contact with the degradation solution. However, CS/Gel/HA scaffolds are more hydrophilic and have loose pore structures, speeding up the degradation process.

Figure 6d shows the mechanical characteristics of scaffolds with different proportion. The compression modulus of CS/Gel/HA scaffolds was 4.1656 ± 0.14 MPa. The compression modulus for scaffolds with a GR content of 0.024, 0.06, and 0.1% were 5.692 ± 0.1305, 7.58973 ± 0.142, and 5.92363 ± 0.13986 MPa, respectively. These results indicate that the addition of trace amounts of GR can significantly enhance the compression modulus of scaffolds. The scaffold with 0.06% GR content had the highest compression modulus, indicating that the mechanical properties of the scaffold does not lineally increase when GR content increased. There is an appropriate range of GR that should be added to enhance the mechanical properties of the scaffold. When the GR content is too high, the scaffold may even become brittle and prone to collapse.

### 3.4. Biological Activity of Bio-Scaffolds

To investigate the cytocompatibility of scaffolds. The distribution and activity of the cells after inoculation on scaffolds printed/produced using different preparations of bio-ink, and culture for 4 days was assessed (Figure 7a). For all scaffolds, BMSCs grew well on the scaffolds, with a high population of living cells and fewer dead cells. Fluorescence microscopy also demonstrates the presence of cells in the different layers of the scaffold, suggesting that the pore size/porosity of these scaffolds supported the proliferation and distribution of these cells. Interestingly, scaffolds doped with GR had more living cells compared to Cs/Gel/HA scaffolds.

In order to further assess the biological activity of cells cultured on the scaffolds, the microstructure of cell-scaffold complexes cultured for 11 days was observed using SEM, under 400, 800 and 1600× magnification (Figure 7b). BMSCs were found to grow and distribute well along the different scaffolds. However, BMSCs were found to be better in Cs/Gel/HA/GR scaffolds compared to the Cs/Gel/HA scaffold, suggesting that small amounts of GR may support cell proliferation due to its hydrophobic properties that improve cell adhesion. GR is hydrophobic and scaffolds doped with GR will provide more attachment sites for cells, promoting cell proliferation and differentiation on the scaffold.

## 4. Conclusions

Cs/Gel and Cs/Gel/HA bio-inks were prepared successfully. The rheological properties for each prepared bio-ink were assessed to identify which bio-ink would be appropriate for 3D bio-printing. Cs:Gel:HA = 1:8:0.02 bio-ink was the most suitable for printing. GR was added to enhance the mechanical properties of the scaffolds. There was no significant effect on the microstructure of scaffolds and they were uniformly dispersed. After further optimization, a Cs/Gel/HA/GR ratio of 1:8:0.02:0.06 was found to have even better water absorption, porosity, compression modulus, and cytocompatibility. Among the composite scaffolds of various proportions, the scaffolds with GR content of 0.06% are the most conducive to cell growth, and the survival number and proliferation of cells are much higher than other scaffolds. This study provides the foundation for further research on tissue engineering of 3D printed cartilage scaffolds for cartilage repair.

## Figures and Tables

**Figure 1 polymers-11-01601-f001:**
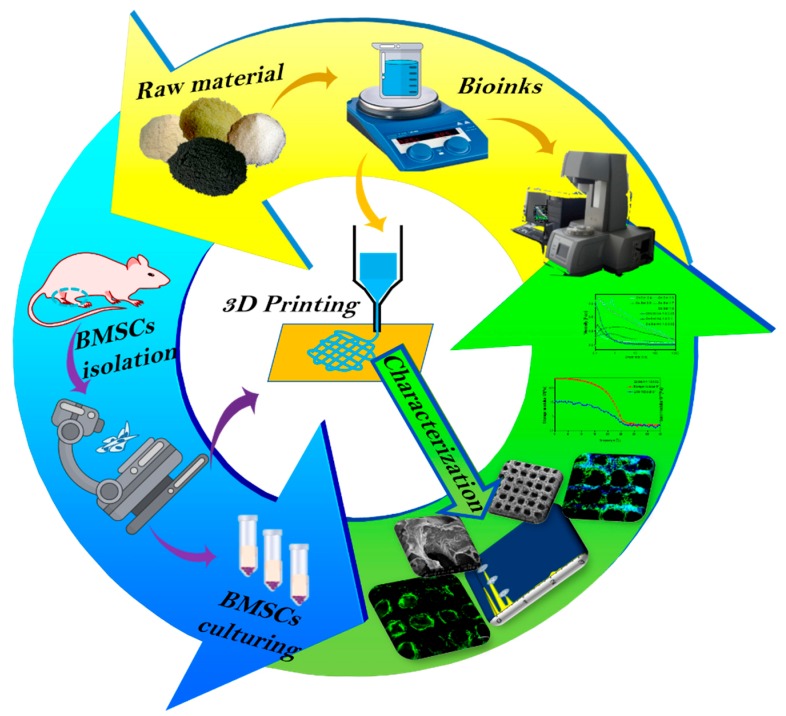
Schematic illustration of the general study design. Yellow area: Preparation of Cs/Gel/HA/GR bio-inks and production of scaffolds using 3D bio-printing. Green area: Characterizations of Cs/Gel/HA/GR cartilage scaffolds using scanning electron microscope (SEM) and live/dead staining of cells.

**Figure 2 polymers-11-01601-f002:**
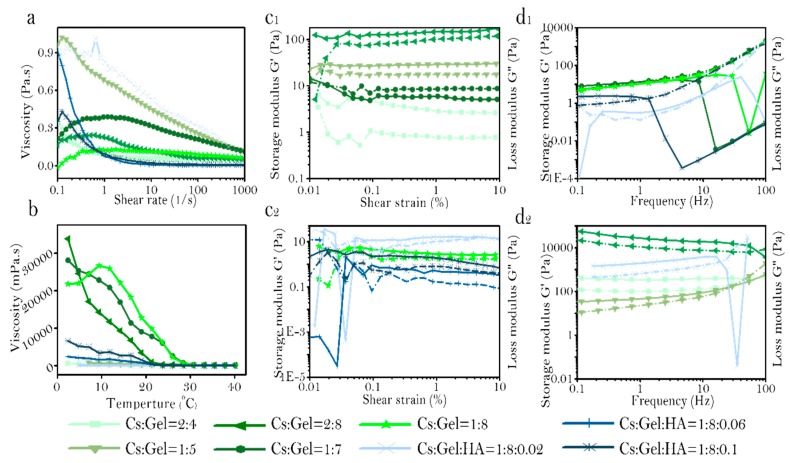
Rheological properties of bio-ink. (**a**): Shear rate, (**b**): Viscosity profile, (**c_1_**,**c_2_**): Strain, and (**d_1_**,**d_2_**): Frequency.

**Figure 3 polymers-11-01601-f003:**
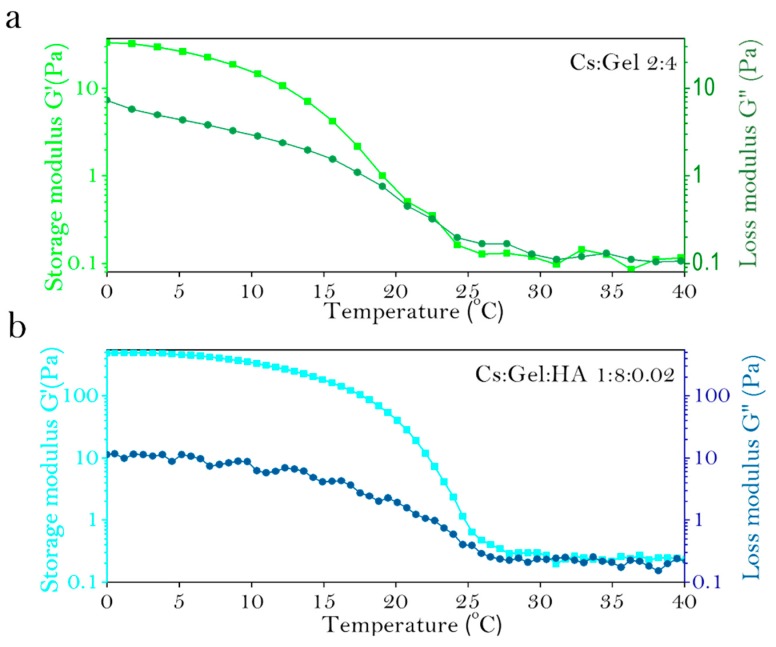
Gel point for various preparations of Cs:Gel:HA bio-ink, (**a**): Cs:Gel = 2:4, (**b**): Cs:Gel:HA = 1:8:0.02.

**Figure 4 polymers-11-01601-f004:**
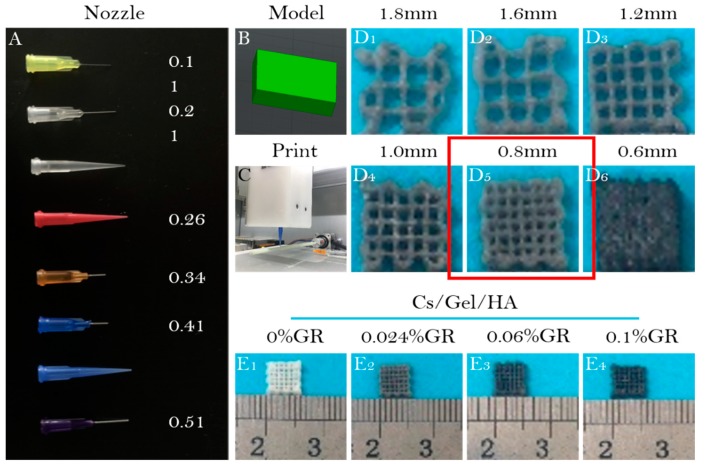
3D bio-printing using bio-ink. (**A**): Needles with varying diameters; (**B**): 3D printing model of bio-scaffold; (**C**): Images of the 3D printer; (**D_1_**–**D_6_**): Appearance of bio-scaffolds with different filling distances at (**D_1_**) 1.8 mm, (**D_2_**) 1.6 mm, (**D_3_**) 1.2 mm, (**D_4_**) 1.0 mm, (**D_5_**) 0.8 mm, and (**D_6_**) 0.6 mm; Appearance of (**E_1_**) Cs/Gel/HA composite scaffold, (**E_2_**) Cs/Gel/HA/0.024%GR composite scaffold, (**E_3_**) Cs/Gel/HA/0.06%GR composite scaffold, and (**E_4_**) Cs/Gel/HA/0.1%GR composite scaffold.

**Figure 5 polymers-11-01601-f005:**
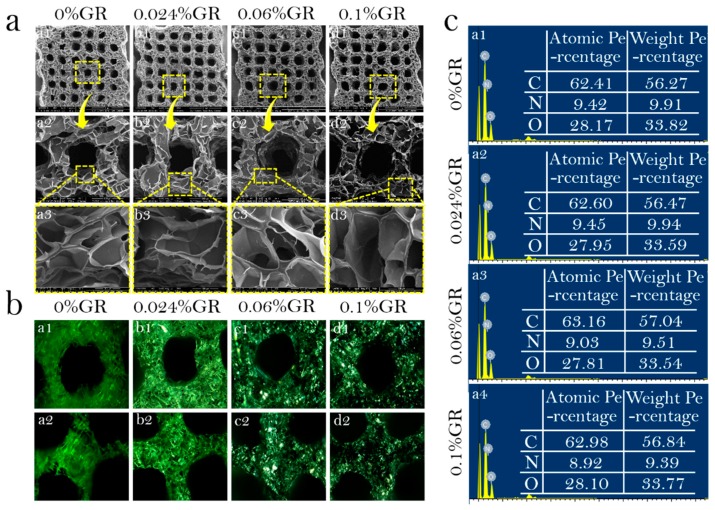
Physical and chemical properties of bio-scaffolds. (**a**): SEM of bio-scaffolds; (**a_1_**–**a_3_**) 0%GR scaffold, (**b_1_**–**b_3_**) 0.024%GR scaffold, (**c_1_**–**c_3_**) 0.06%GR scaffold, (**d_1_**–**d_3_**) 0.1%GR scaffold, (**a_1_**–**d_1_**) 50×, (**a_2_**–**d_2_**) 200×, (**a_3_**–**d_3_**) 800×; (**b**): Polarized microscope of bio-scaffolds. (**a_1_**,**a_2_**) 0%GR scaffold, (**b_1_**,**b_2_**) 0.024%GR scaffold, (**c_1_**,**c_2_**) 0.06%GR scaffold, (**d_1_**,**d_2_**) 0.1%GR scaffold; (**c**): EDS of bio-scaffolds. (**a_1_**) 0%GR scaffold, (**a_2_**) 0.024%GR scaffold, (**a_3_**) 0.06%GR scaffold, (**a_4_**) 0.1%GR scaffold.

**Figure 6 polymers-11-01601-f006:**
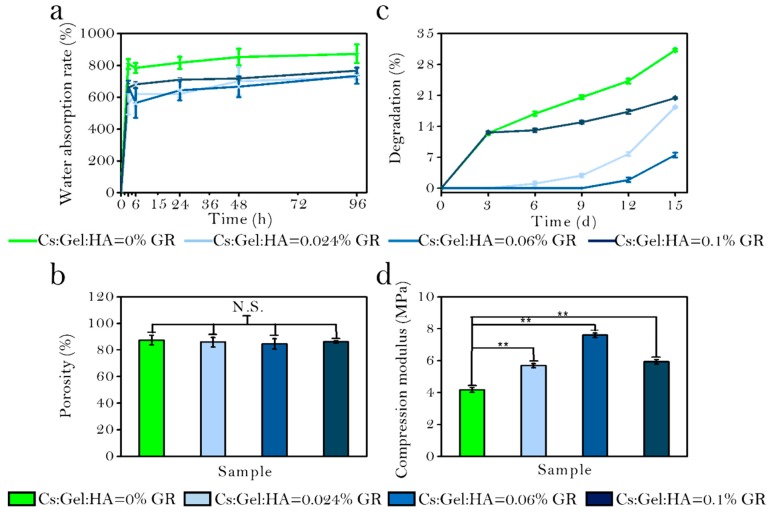
(**a**): Water absorption rate of bio-scaffolds; (**b**): Porosity of bio-scaffolds; (**c**): Degradation rate of bio-scaffolds; (**d**): Mechanical strength of bio-scaffolds; *** *p* < 0.001; ** *p* < 0.01; ** *p* < 0.05; N.S.: *p* ≥ 0.05: no significant difference.

**Figure 7 polymers-11-01601-f007:**
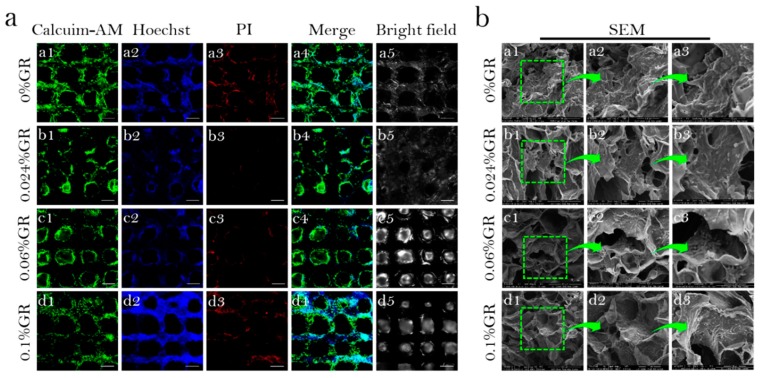
Biological activity of bio-scaffolds. (**a**): Viability and distribution of BMSCs on CS/Gel/HA/GR scaffolds. (**a_1_**–**a_5_**) 0% GR scaffold, (**b_1_**–**b_5_**) 0.024% GR scaffold, (**c_1_**–**c_5_**) 0.06% GR scaffold, (**d_1_**–**d_5_**) 0.1% GR scaffold, (**a_1_**–**d_1_**): Calcuim-AM stain of CS/Gel/HA/GR scaffolds, (**a_2_**–**d_2_**): Hoechst stain of CS/Gel/HA/GR scaffolds, (**a_3_**–**d_3_**): PI stain of CS/Gel/HA/GR scaffolds, (**a_4_**–**d_4_**): Merge stain of CS/Gel/HA/GR scaffolds, (**a_5_**–**d_5_**): Bright field of CS/Gel/HA/GR scaffolds, (Scale: 50 µm); (**b**): SEM of bioscaffold-cell composite. (**a_1_**–**a_3_**) 0% GR scaffold, (**b_1_**–**b_3_**) 0.024%GR scaffold, (**c_1_**–**c_3_**) 0.06% GR scaffold, (**d_1_**–**d_3_**) 0.1% GR scaffold, (**a_1_**–**d_1_**) 400×, (**a_2_**–**d_2_**) 800×, (**a_3_**–**d_3_**) 1600×.

**Table 1 polymers-11-01601-t001:** Composition ratio of different bio-ink.

Sample	CS (*w*/*v* %)	Gel (*w*/*v* %)	HA (*w*/*v* %)	GR (*w*/*v* %)
1	2	4	-	-
2	1	5	-	-
3	2	6	-	-
4	1	7	-	-
5	1	8	-	-
6	1	8	0.02	-
7	1	8	0.06	-
8	1	8	0.1	-
9	1	8	0.02	0.024
10	1	8	0.02	0.06
11	1	8	0.02	0.1

**Table 2 polymers-11-01601-t002:** Print parameter conditions for various ratios of bio-ink.

-	GR: 0%	GR: 0.024%	GR: 0.06% & 0.1%
Nozzle temperature (°C)	14	18	18
Platform temperature (°C)	2	2	2
Print speed (mm/s)	2	4	4
Probe stress	From 0.08 MPa, each increase of 0.02 MPa gradually increases	From 0.05 MPa, each increase of 0.01 MPa gradually increases	From 0.05 MPa, each increase of 0.01 MPa gradually increases
Print thick (mm)	0.2	0.2	0.2
Print height (mm)	2	2	2
Print layer	10	10	10
Filling distance (mm)	0.6–0.8	0.6–0.8	0.6–0.8
Probe type	0.21 mm needle	0.21 mm tapered	0.21 mm tapered

**Table 3 polymers-11-01601-t003:** Linear viscoelastic regions of various ratios of bio-ink.

Sample	Linear Viscoelastic Range (%)
Cs:Gel = 2:4	0.06–3
Cs:Gel = 1:5	0.03–10
Cs:Gel = 2:6	0.03–10
Cs:Gel = 1:7	0.2–10
Cs:Gel = 1:8	0.2–9
Cs:Gel:HA = 1:8:0.02	0.3–2
Cs:Gel:HA = 1:8:0.06	0.8–10
Cs:Gel:HA = 1:8:0.1	0.5–2

**Table 4 polymers-11-01601-t004:** Fluid behavior of various ratios of bio-ink.

Sample	Strain (%)	0 Hz	0.1–1 Hz	1–10 Hz	Summary
Cs:Gel = 2:4	1	Gelatin	Solid-like	Solid-like	Well
Cs:Gel = 1:5	1	Gelatin	Solid-like	Solid-like	Well
Cs:Gel = 2:6	1	Gelatin	Solid-like	Solid-like	Well
Cs:Gel = 1:7	2	Viscoelastic solid	Solid-like	Fluid-like	Bad
Cs:Gel = 1:8	2	Viscoelastic solid	Solid-like	Fluid-like	Bad
Cs:Gel:HA = 1:8:0.02	0.5	Gelatin	Solid-like	Solid-like	Well
Cs:Gel:HA = 1:8:0.06	2	Viscoelastic liquid	Solid-like	Solid-like	Bad
Cs:Gel:HA = 1:8:0.1	0.8	Viscoelastic solid	Solid-like	Fluid-like	Bad

**Table 5 polymers-11-01601-t005:** Summary of fluid properties of various ratios of bio-ink.

Sample	Fluid Characteristic	Linear Viscoelastic Range (%)	FREQUENCY Scanning	Sticky Temperature Curve	Comment
Cs:Gel = 2:4	Well	0.06–3	Well	Low	Well
Cs:Gel = 1:5	-	0.03–10	Well	Low	-
Cs:Gel = 2:6	-	0.03–10	Well	High	-
Cs:Gel = 1:7	-	0.2–10	-	High	-
Cs:Gel = 1:8	-	0.2–9	-	High	-
Cs:Gel:HA = 1:8:0.02	Well	0.3–2	Well	Low	Well
Cs:Gel:HA = 1:8:0.06	-	0.8–10	-	High	-
Cs:Gel:HA = 1:8:0.1	Well	0.5–2	-	High	-

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
