# Peer review of "3D Bio-Printing of CS/Gel/HA/Gr Hybrid Osteochondral Scaffolds"

_polymers, 2019, doi:10.3390/polym11101601_

Round 1

Reviewer 1 Report

Line 92 - The authors should better explain which solvent they used to make the Cs/Gels/HA/GR  solution.

Line 93 - The authors mixed CS, gelatin, hyaluronic acid. Gelatin is a pH sensitive polymer, the authors must indicate which was the working pH.

Line 93 - Graphene is not soluble in water, the author should explain how they prepare a GR solution.

Line 94 - It is not clear if the scaffold s are made of Cs/Gels/HA/GR or each single polymer made a scaffold. Please explain.

Line 96 - If BMSCs were seeded onto the scaffold, which was the advantage of 3D-bioprinting? Usually  cells are mixed with the bioink and printed.

Line 109 - Gelatinn and  hyalyuronic acid should be added in the Materials section.

Line 121 - carbodiimide / N-hydroxy-succinamide / morpholine ethane sulfonate  should be reported in the Materials section.

Reviewer 2 Report

Dear Authors

This manuscript is quite good for the future researchers to work, could you please follow my suggestions for the betterment of the paper;

1) Improve your Scientific English

2) Improve the Introduction and Conclusion linked it with an experiment. 

Other is fine 
